# Factors Associated with Risk of Diabulimia in Greek Adult Population with Type 1 Diabetes Mellitus

Kiriaki Apergi [1,*], Maria Romanidou [2], Hesham Abdelkhalek [2] and Fragiskos Gonidakis [3]

[1] 1st Department of Propaedeutic and Internal Medicine, Laiko General Hospital, Medical School, National and Kapodistrian University of Athens, Mikras Asias 75, Goudi, 11527 Athens, Greece

[2] Essex Partnership University NHS Foundation Trust, Lodge, Runwell, Wickford SS11 7XX, UK

[3] 1st Department of Psychiatry, Eginition Hospital, Medical School, National and Kapodistrian University of Athens, Vasilissis Sophias 72, 11528 Athens, Greece

* Correspondence: kiapergi@med.uoa.gr

**Abstract:** Background: Diabulimia is associated with "resistance to treatment", impaired glycemic control, and increased risk of diabetic complications. The aim of this study was to explore the possible lifestyle and diet factors associated with diabulimia using the "Diabetes Eating Problem Survey-Revised" (DEPS-R), a questionnaire assessing the risk of diabulimia in patients with T1DM. Methods: 100 outpatients with a diagnosis of T1DM for over a year, from two hospitals in Athens, Greece were asked to complete a questionnaire about their medical history and lifestyle behaviors along with the Eating Attitudes Test (EAT-26), the Beck Depression Scale (BDI-II) and the DEPS-R. Results: Multivariate linear regression analysis showed statistically significant associations between DEPS-R score and HbA1c% levels (b = 4.447, 95% CI 3.220–5.675), sex (male) (b = −4.413, 95% CI −8.778–−0.047, weight perception higher than normal (b = 9.896, 95% CI 5.266–14.526), time spent walking minutes per week (b= −0.019, 95% CI −0.031–−0.006), having received diabetes nutritional education (b = −5.128, 95%CI −10.141–−0.115), eating breakfast (b = −6.441, 95% CI −11.047–−1.835) and having a first relative diagnosed with an eating disorder (b = 6.164, 95%CI 0.030–12.298). The presence of these factors could help highlight the profile of potential individuals at risk of diabulimia and enhance preventive interventions.

**Keywords:** diabulimia; eating disorders; diabetes mellitus; diabetes mellitus type 1

## 1. Introduction

Type 1 diabetes (T1DM) accounts for 5–10% of patients with diabetes mellitus [1] and it is believed to be a multifactorial disease with a contribution from genetic and environmental factors resulting in autoimmune reaction, usually leading to absolute insulin deficiency [2]. Optimized glycemic control in individuals with T1DM can significantly delay the onset and progression of severe complications of diabetes [3]. Insulin, along with lifestyle management and psychosocial care, are the cornerstones of T1DM management [2]. According to American Diabetes Association, all T1DM patients should be referred for diabetes self-management education and support, medical nutrition therapy, and assessment of psychosocial/emotional health concerns. An individualized healthy eating pattern, along with regular physical activity and proper insulin therapy, has been proven to be the key components of T1DM management for the prevention or delay of diabetes complications [2,3].

Health outcomes for T1DM individuals are usually highly dependent on the patients' ongoing self-care behavior and for this reason, it is considered a psychologically challenging chronic condition [4]. Several determinants have been found to affect patient adherence to T1DM therapy, such as individual, social, and other environmental factors [5]. Additionally, psychiatric comorbidities, including eating disorders and especially diabulimia, are currently recognized as a major factor leading to "resistance to treatment" and inability

to achieve the glycemic targets [5]. Despite the advances in diabetes treatment, 60% of T1DM individuals admit to misusing insulin [6]. Since the publication of the first article reported the misuse of insulin in T1DM individuals in 1983 [7], the scientific and medical community has shown considerable interest in diabulimia, an eating disorder characterized by restraining or omitting insulin in order to control body weight, and its influence on T1DM management. Although diabulimia is not yet recognized as a mental illness by the Diagnostic and Statistical Manual of Mental Disorders (DSM-V) [8,9], because of its impact on health outcomes [10], many diabetes associations now propose that screening and monitoring of eating disorders should be an integral part of diabetes care [4]. If left untreated, diabulimia as defined by the voluntary restriction of insulin in order to lose weight may lead to serious consequences, both acute and long-term, which are similar to those of inadequately treated diabetes. Acute consequences could include repeated crises of ketoacidosis. In the long-term, diabulimia might lead to classic diabetes complications earlier in life affecting multiple organ systems, such as the kidneys, nervous system, cardiovascular system, and retina [6,8]. Hence, the purpose of the study was to research possible factors related to diabulimia as measured by the score in Diabetes Eating Problem Survey-revised (DEPS-R) [11–13], a questionnaire specifically designed for the assessment of diabulimia risk in adults with T1DM.

## 2. Materials and Methods

This cross-sectional study was conducted between April 2018 to September 2018 at the outpatient diabetes clinics in "Attikon" and "Laiko" general hospitals in Athens, Greece. Ethical approval was granted by the Ethics Committee of the "Attikon" General University Hospital, Athens, Greece (approval code: ΒΠΠΚ ΕΒΔ245/ 20 April 2018). All procedures were in line with the Declaration of Helsinki [14]. More details about the study design can be found elsewhere [13]. In brief, 140 T1DM outpatients aged 18- to 70-year-olds with a formal diagnosis of T1DM for more than one year were approached to participate in the study during their regularly scheduled medical visits. Of the approached patients, 100 gave informed consent and completed the study questionnaires at the study site in presence of the investigator or via an online electronic form.

### 2.1. Inclusion Criteria

The inclusion criteria were: (1) capacity and willingness to provide informed consent, (2) age 18–70 years, (3) formal diagnosis of T1DM for more than one year.

### 2.2. Exclusion Criteria

The exclusion criteria were: missing data for any of the variables.

Participants were asked to complete the self-administered questionnaire which included: demographic data (education, family status and living status), questions regarding weight attitudes (frequency of weighing per week, perception of their weight in comparison to ideal weight, weight change during the previous 6 months, attempts for weight loss), diabetes self-management (number of daily self-blood glucose monitoring, continuous glucose monitoring use, number of hypoglycemic events per week, total insulin units per day), exercise (weekly time spent in minutes performing organized exercise eg. at the gym, swimming etc., weekly time spent in minutes walking other than for organized exercise), diabetes nutrition management [having received diabetes nutritional education once in their life, carbohydrates (carbs) counting method (carbs exchanges, carb grams, both ways vs no counting), tools for carb counting (approximately by sight, cups and spoons, scale, no measure), eating/ skipping breakfast, having regular meals/snacks or snacking all day/eating large portions without a meal plan all day, being under dietitian's supervision-counselling, being on diet], medical history (first relative diagnosis of an Eating Disorder).

In addition to the above, participants were asked to complete: the Eating Attitudes Test questionnaire (EAT-26) [15], the DEPS-R questionnaire [13], and the Beck Depression Inventory questionnaire (BDI-II) [16,17]. Biochemical biomarker (HbA1c%) and prescribed treatment were

collected from the patient's folder. The height was measured to the nearest centimeter using a stadiometer, and the weight was measured (unclothed) to the nearest 0.1 kg using a calibrated balance scale. For the online questionnaires, weight and height were self-reported. Body mass index (BMI) was calculated using the weight (kg)/height (m$^2$) equation.

### 2.3. Study Questionnaires

DEPS-R is a self-administered scale designed to measure disordered eating behaviors in patients with diabetes. The original 28-item DEPS was revised and shortened into the DEPS-R by Markowitz et al. [12]. The DEPS-R is composed of 16 items on a 6-point Likert (0 = never; 1 = rarely; 2 = sometimes; 3 = often; 4 = usually; 5 = always). A recommended cut-off score of ≥20 has been empirically established as a threshold indicating the need for further clinical assessment of the eating pathology [12]. The questionnaire was translated into Greek [13] in line with the standardized procedure suggested by Brislin [18–20].

The EAT-26 is a 26-item questionnaire developed by Garner et al. [15], and it is widely used as a standardized self-report measure of disordered eating behaviors [21–23]. In the Greek version of the questionnaire, a score ≥20 is considered to be a high risk of the presence of disordered eating [24].

The Beck Depression Inventory (BDI, BDI-1A, BDI-II), is a self-report 21-item questionnaire developed by Aaron T. Beck. It can be used as a psychometric test for measuring the severity of depression [16,17]. According to previous studies in T1DM patients from Greece, a total score ≥14 points is considered as indicative of depressive symptoms [25].

### 2.4. Statistical Analysis

The analyses included only participants with complete data. Total scores were calculated for all study questionnaires. Descriptive statistics were provided as mean and standard deviation for continuous variables and relative frequencies for categorical variables. The normality assumption was based on the Shapiro-Wilk test and histograms [26]. Univariate analyses were performed for every variable. In order to assess the relationships between the dependent variable (DEPS-R score) and the covariates and conclude to the final model, a stepwise multivariable linear regression, including all variables with a *p*-value < 0.2 in univariate analyses, was performed. The estimated b- coefficients, (95% confidence intervals (CI) were used to describe these associations. The level of significance was defined as a *p*-value < 0.05.

Statistical analyses were conducted using RStudio Team (2020) [27].

## 3. Results

The demographic, lifestyle, and clinical characteristics of the sample (N = 100) are displayed in Table 1. Of the 100 participants, 29% were males. No patients with late-onset autoimmune diabetes in adults (LADA) were identified in the current sample. The mean age of the participants was 35.9 (±10.7) years, the mean BMI was 24.6 (±5.0) kg/m$^2$ and the mean HbA1c level was 7.6 (±1.7). 81% of the participants had received nutritional education for T1DM management once in their life, 75% of the participants believed that their weight was higher than normal, 56% were trying to lose weight and 20% had nutrition therapy receiving dietetic consultation for at least once after their diagnosis. The mean scores for EAT-26 were 17.3 (±10.4), BDI-II 10.9 (±10.0), and DEPS-R 19.2 (±14.7).

The univariate regression analysis (Table 2) showed a positive correlation between DEPS-R score and HbA1c %, higher than normal weight perception, weight loss over the previous six months, attempts to lose weight, having a first relative diagnosis of an Eating Disorder, EAT-26 score, and BDI-II. On the contrary, being male, time spent performing organized exercise at the gym/sports (min/week), time spent walking (min/week), having received diabetes nutritional education, eating breakfast, and having regular meals/snacks were inversely correlated with the DEPS-R score. No statistically significant interaction was found between the examined variables with sex or age.

**Table 1.** Descriptive characteristics (N = 100).

|  | Mean | SD | % |
|---|---|---|---|
| **Age (years)** | 35.9 | 10.7 |  |
| **Sex (males %)** |  |  | 29.0 |
| **Education status (years)** | 14.7 | 2.5 |  |
| **Living Status** |  |  |  |
| With parents % |  |  | 31.0 |
| With housemate % |  |  | 3.0 |
| With companion/spouse % |  |  | 48.0 |
| With relatives % |  |  | 3.0 |
| Alone % |  |  | 15.0 |
| **Family Status** |  |  |  |
| Unmarried % |  |  | 32.0 |
| Unmarried in relationship % |  |  | 21.0 |
| Unmarried with child(ren) % |  |  | 3.0 |
| Married/no child(ren) % |  |  | 13.0 |
| Married/with child(ren) % |  |  | 26.0 |
| Divorced/widow % |  |  | 5.0 |
| **BMI (kg/m$^2$)** | 24.6 | 5.0 |  |
| **Weighing (times/week)** | 3.97 | 5.9 |  |
| **Weight perception** |  |  |  |
| **Higher than normal %** |  |  | 75.0 |
| **Weight Change during the last 6 months** |  |  |  |
| Stable ($\pm$2 kg) % |  |  | 51.0 |
| Weight loss % |  |  | 23.0 |
| Weight Gain % |  |  | 26.0 |
| **Duration of diabetes (years)** | 17.0 | 10.1 |  |
| **Type of insulin administration (Insulin Pump Users %)** |  |  | 65.0 |
| **CGM users %** |  |  | 56.0 |
| **SMBG frequency (checks/day)** | 5.4 | 3.2 |  |
| **Hypoglycemia events (episodes/week)** | 2.13 | 3.1 |  |
| **Organized Exercise (min/week)** | 106.5 | 14.8 |  |
| **Walking (min/week)** | 134.8 | 16.2 |  |
| **Diabetes nutritional education %** |  |  | 81.0 |
| **Carb counting method** |  |  |  |
| No counting % |  |  | 27 |
| Carb exchanges % |  |  | 36 |
| Carb grams % |  |  | 33 |
| Both ways % |  |  | 4 |
| **Tools for carb counting** |  |  |  |
| No measure % |  |  | 14 |
| Approximately by sight % |  |  | 49 |
| Cups and spoons % |  |  | 9 |
| Scale % |  |  | 28 |
| **Eating Breakfast %** |  |  | 74.0 |
| **Having regular meals/snacks %** |  |  | 92.0 |
| **Try to lose weight %** |  |  | 56.0 |
| **Dieting %** |  |  | 36.0 |
| **Being Under Dietitian's Supervision %** |  |  | 20.0 |
| **First relative diagnosed with eating disorder %** |  |  | 13.0 |
| **EAT-26 score** | 17.3 | 10.4 |  |
| **BDI-II** | 10.9 | 10.0 |  |
| **DEPS-R** | 19.2 | 14.7 |  |

S.D: Standard Deviation, BMI: Body Mass Index, HbA1c%: Hemoglobin A1c, CGM: Continuous Glucose Monitor, SMBG: Self-Monitoring Blood Glucose, EAT-26: Eating Attitudes Test, BDI-II: Beck Depression Inventory-II, DEPS-R: Diabetes Eating Problem Survey-Revised.

**Table 2.** Factors correlating with DEPS-R score based on Univariate Linear Regression models.

| Factor | Unstandardised b Coefficient | 95% CI | df | Overall F Test | Overall *p*-Value | Adjusted R-Squared |
|---|---|---|---|---|---|---|
| **HbA1c %** | **5.168** | **3.759, 6.576** | **1, 98** | **52.98** | **<0.001** | **0.344** |
| **Age** (years) | 19.90 | −0.042, 0.499 | 1, 98 | 2.811 | 0.097 | 0.018 |
| **Sex** (males vs females) | **−8.226** | **−14.453, −1.999** | **1, 98** | **6.873** | **0.010** | **0.056** |
| **Education** (years) | −0.5627 | −1.728, 0.602 | 1, 98 | 0.9187 | 0.340 | 0.001 |
| **Living Status** | | | 4, 95 | 0.8502 | 0.497 | 0.001 |
| With housemate | −6.054 | −23.698, 11.591 | | | | |
| With companion/spouse | 1.759 | −4.965, 8.483 | | | | |
| With relatives | 10.280 | 7.365, 27.924 | | | | |
| Alone | 5.746 | −3.432, 14.925 | | | | |
| **Family status** | | | 5, 94 | 0.2559 | 0.936 | 0.004 |
| Unmarried in relationship | −0.353 | −8.682, 7.977 | | | | |
| Unmarried with child(ren) | 3.552 | −14.357, 21.461 | | | | |
| Married/no child(ren) | −1.012 | −10.767, 8.743 | | | | |
| Married/with child(ren) | −2.589 | −10.420, 5.242 | | | | |
| Divorced/widow | 4.219 | −10.044, 18.482 | | | | |
| **BMI** (kg/m$^2$) | 0.402 | −0.178, 0.982 | 1, 98 | 1.890 | 0.172 | 0.009 |
| **Weighing**(times/week) | 0.087 | −0.409, 0.583 | 1, 98 | 0.122 | 0.728 | 0.008 |
| **Weight perception** Higher than normal | **8.613** | **2.088, 15.139** | **1, 98** | **6.861** | **0.010** | **0.056** |
| **Weight Change (last 6 months)** | | | **2, 97** | **4.133** | **0.019** | **0.060** |
| Weight loss | **9.834** | **2.749, 16.918** | | | | |
| Weight Gain | 5.722 | −1.075, 12.519 | | | | |
| **Duration of diabetes** (years) | −0.045 | −0.338, 0.248 | 1, 98 | 0.093 | 0.761 | 0.009 |
| **Insulin administration** | | | | | | |
| Insulin pump | −0.998 | −7.123, 5.127 | 1, 98 | 0.105 | 0.747 | 0.009 |
| **CGM use** | −0.204 | −0.589, 0.181 | 1, 53 | 1.131 | 0.293 | 0.002 |
| **SMBG frequency (checks/day)** | −0.676 | −1.590, 0.238 | 1, 45 | 2.156 | 0.145 | 0.012 |

**Table 2.** *Cont.*

| Factor | Unstandardised b Coefficient | 95% CI | df | Overall F Test | Overall p-Value | Adjusted R-Squared |
|---|---|---|---|---|---|---|
| **Hypoglycemic events (episodes/week)** | 0.300 | −5.666, 6.266 | 1, 98 | 0.009 | 0.921 | 0.001 |
| **Organized Exercise (min/week)** | **−0.023** | **−0.043, −0.003** | **1, 98** | **5.056** | **0.027** | **0.040** |
| **Walking (min/week)** | **−0.023** | **−0.041, −0.005** | **1, 98** | **6.262** | **0.014** | **0.051** |
| **Diabetes nutritional education** | **−9.575** | **−16.600, −2.550** | **1, 98** | **7.317** | **0.008** | **0.060** |
| **Carb counting method** | | | 3, 96 | 1.617 | 0.191 | 0.018 |
| Carb exchanges | −6.213 | −13.551, 1.125 | | | | |
| Carb grams | −5.609 | −13.089, 1.870 | | | | |
| Both ways | −13.602 | −29.043, 1.839 | | | | |
| **Tools for carb counting** | | | 3, 96 | 1.762 | 0.160 | 0.023 |
| Approximately by sight | −8.429 | 17.144, 0.286 | | | | |
| Cups and spoons | −9.032 | 21.319, 3.256 | | | | |
| Scale | **−10.643** | **−20.057, −1.229** | | | | |
| **Eating Breakfast** | **−9.838** | **−16.203, −3.473** | **1, 98** | **9.407** | **0.003** | **0.078** |
| **Having regular meals/snacks** | **−13.076** | **−23.527, −2.626** | **1, 98** | **6.166** | **0.015** | **0.049** |
| **Try to lose weight** | **9.687** | **4.128, 15.246** | **1, 98** | **11.96** | **<0.001** | **0.089** |
| **Dieting** | 0.655 | −5.434, 6.743 | 1, 98 | 0.046 | 0.832 | 0.010 |
| **Being Under Dietitian's Supervision %** | 1.350 | −5.952, 8.652 | 1, 98 | 0.135 | 0.715 | 0.010 |
| **First relative diagnosed with Eating Disorder** | **10.534** | **2.103, 18.965** | **1, 98** | **6.148** | **0.015** | **0.050** |
| **EAT-26 score** | **0.537** | **0.278, 0.797** | **1, 98** | **16.85** | **<0.001** | **0.138** |
| **BDI-II** | **0.754** | **0.501, 1.007** | **1, 98** | **34.89** | **<0.001** | **0.255** |

BMI: Body Mass Index, HbA1c%: Hemoglobin A1c, CGM: Continuous Glucose Monitor, SMBG: Self-Monitoring Blood Glucose, EAT-26: Eating Attitudes Test, BDI-II: Beck Depression Inventory-II, DEPS-R: Diabetes Eating Problem Survey-Revised. Bold: Statistic significance <0.05.

Multivariate linear regression analysis (Table 3) showed a statistically significant association between DEPS-R score and HbA1c% levels (b = 4.447, 95% CI 3.220–5.675), sex (male) (b = −4.413, 95% CI −8.778––0.047, weight perception higher than normal (b = 9.896, 95% CI 5.266–14.526), time spent walking per week (b= −0.019, 95% CI −0.031––0.006), having received diabetes nutritional education (b = −5.128, 95%CI −10.141––0.115), eating breakfast (b = −6.441, 95% CI −11.047––1.835) and having a first relative diagnosed with an eating disorder (b = 6.164, 95%CI 0.030–12.298). The overall *p*-value of the multivariable model was statistically significant (*p* < 0.001).

**Table 3.** Factors correlating with DEPS-R score based on multivariate linear regression model.

| Factor | Unstandardised b Coefficient | 95% CI | *p*-Value | df | Overall F Test | Overall *p*-Value | Adjusted R-Squared |
|---|---|---|---|---|---|---|---|
| **HbA1c %** | **4.447** | **3.220, 5.675** | **<0.001** | 7, 92 | 19.01 | **<0.001** | 0.560 |
| **Sex** (males) | **−4.413** | **−8.778, −0.047** | **0.047** | | | | |
| **Weight perception Higher than normal** | **9.896** | **5.266, 14.526** | **<0.001** | | | | |
| **Walking** (min/week) | **−0.019** | **−0.031, −0.006** | **0.004** | | | | |
| **Diabetes nutritional education** | **−5.128** | **−10.141, −0.115** | **0.045** | | | | |
| **Eating Breakfast** | **−6.441** | **−11.047, −1.835** | **0.007** | | | | |
| **First relative diagnosed with Eating Disorder** | **6.164** | **0.030, 12.298** | **0.048** | | | | |

HbA1c%: Hemoglobin A1c, Bold: Statistically significant <0.05.

## 4. Discussion

This study shows that HbA1c levels, female sex, patients perceiving their body weight to be higher than normal, not having received adequate diabetes nutritional education, skipping breakfast, and having a first relative diagnosis of an eating disorder were factors positively associated with DEPS-R score, a questionnaire designed and validated to assess the high risk for diabulimia [11–13]. Factors such as weight change over the previous six months, attempts to lose weight, performing organized exercise, having regular meals/snacks, EAT-26 score, and BDI-II were positively correlated to DESP-R score in univariate analysis but lost their statistical significance in the multivariable analysis. These results highlight the factors which the medical team should pay special attention to for an earlier diagnosis, risk modification, and management of diabulimia.

In terms of both predisposed risk factors, morbidity, and comorbidity the association of T1DM and eating disorders, and specifically diabulimia, is complex and still has not been adequately addressed. T1DM may offer a fertile ground for the development of eating disorders. The patients with T1DM need to be constantly aware of food composition, especially in terms of carbohydrate context [4] and sometimes they may follow a diet with many dietary restrictions [6,8]. This behavior and constant occupation with food and diet rules could potentially stimulate a variety of diverse kinds of reactions, including feelings of guilt every time when any dietary rules are not strictly followed. This situation is quite similar to the cognitive patterns that are recognized in eating disorders [6,7]. On the other hand, medical nutrition therapy for diabetes is essential for the optimal management of both T1DM and type 2 diabetes in adults [28,29]. Guidelines and recommendations for diabetes medical care for optimal diabetes management and complications' prevention, as derived by numerous professional and health organizations worldwide, have constantly highlighted the importance of receiving diabetes nutrition therapy and nutritional counseling as the cornerstone for effective diabetes care management, not only initially with the diagnosis, but also ongoing along with each individual's patient therapeutic plan [28–30]. Along with these guidelines, the current study also showed that having received diabetes nutrition

education at least once following the diagnosis of T1DM was inversely associated with the DEPS-R score. It is true that for the majority of patients with diabetes mellitus, the most challenging part of the treatment plan is determining what to eat. The latest guidelines advise that there is not a "one-size-fits-all" healthy eating pattern [28]. For this reason, meal planning should be individualized for each patient with T1DM. This also includes active engagement with patients with T1DM through education, promoting self-management, and collaborative treatment planning with the specialist diabetes team. Thus, medical management and diabetes nutritional education demonstrate an important role in overall diabetes management, assisting individuals with T1DM to adjust their diet to personal needs, tastes, and social life, clarifying common diet myths, and avoiding unnecessary diet restrictions [28,29]. Data from previous research, confirm the significance of nutritional therapy delivered by registered dietitians. Individuals with T1DM that had received nutritional therapy presented absolute decreases in HbA1c up to 1.9% both at three- and six-months intervals [30].

Having regular breakfast was associated with decreased DEPS-R score. On the contrary, having longer periods of fasting, snacking multiple times during the day, or eating large quantities without a meal plan was associated with an increased DEPS-R score. Other studies showed that having a regular eating pattern, including consumption of breakfast and multiple smaller meals during the day, was associated with better glycemic control in adults with T1DM [5,31]. Also, eating more frequently, having breakfast, and consuming three meals every day, was an important clinical application for the management of binge eating disorder [32]. It is therefore important for the patients with T1DM to receive nutritional education including regular meals, not only with a view to optimize glycemic control, but with an additional focus to decrease the risk of disordered eating behaviours.

Sex is found to be associated with the risk of developing several mental health conditions, including depression, anxiety, and eating disorders. Females are usually more susceptible to developing such conditions [6,8,33,34] and 30 to 40% of the females with T1DM, have disordered eating behaviors compared to 9–11% of males [35]. The current study comes in agreement with these results. The female participants of the study showed to have increased DEPS-R score by approximately 4 units compared to males after adjusting to all the other variables.

Males and females with disordered eating or an eating disorder diagnosis seem to experience body image issues and increased concerns about their body weight [36,37]. Specifically, the early emergence of body dissatisfaction has been unsurprisingly linked to the development of eating disorders, including anorexia nervosa and bulimia nervosa in childhood and adolescence [15,38]. Moreover, individuals with T1DM are prone to the development of a negative body image because of dietary restrictions, weight variations, perception of living in an unhealthy body, and focus and attention on the body [39]. However, in a meta-analysis, the findings varied across studies as 3 studies reported mixed results, 17 studies found that body dissatisfaction was common and that body concerns were generally greater in youth with T1DM compared to controls, 9 studies did not find any differences in body image dissatisfaction between participants with and without T1DM and 3 studies described higher body satisfaction in youth with T1DM than in controls. This variability could be explained by the variation of the assessment tools across studies, the overrepresentation of female subjects, and the cross-sectional type of analysis conducted [39]. Nevertheless, individually addressing the issue of a negative body image in T1DM patients could be a helpful practice in order to decrease the risk of developing diabulimia.

In the univariate model, EAT-26 was positively correlated with DEPS-R, but lost significance when controlled for covariates. This could be explained by the fact that the traditional questionnaires for eating disorders are not so effective when applied to diabetes patients [11,40–42]. Also, there is an increased prevalence of depression and anxiety in individuals with T1DM compared to healthy controls with positive correlations between poor metabolic control [43,44]. However, in the current study the risk for diabulimia, as

assessed by the DEPS-R score, depended on the risk for depression and anxiety disorders as indicated by BDI-II, only in the univariate analysis. In the multivariable analysis, the effect of the BDI-II score was not significant which reflects that the effect was mediated by the covariates included in the final model.

In this study, having a first relative diagnosis of an eating disorder was positively associated with the DEPS-R score. Eating disorders are complex psychiatric conditions with a combination of genetic and environmental factors [45]. Individual polygenic scores have been developed to assess the risk and severity of eating disorders and especially anorexia nervosa [45]. To our knowledge, there is no published research associating diabulimia risk with any genetic factors. Given the results of the current analysis and data on the genetic predisposition of eating disorders, it seems plausible that there could be an association that can be explored in future research.

This study was the first attempt to examine the behavioral and lifestyle practice in association with diabulimia risk in the Greek T1DM population. However, this study had some limitations: only HbA1c% was used as a glycemic control factor due to the inability to perform more laboratory tests. Another limitation of this research was that most of the volunteers were females and insulin pump users, which may have introduced a bias. Finally, the majority of data collected was self-reported, and the questions about demographic data, diet intake, and physical activity were collected through common questions and not with the use of validated instruments, e.g., a food frequency questionnaire, which could have added some bias in the estimations assessed in this study.

## 5. Conclusions

It is crucial to evaluate the risk of developing disordered eating and eating disorders specifically diabulimia in T1DM patients of all ages in order to enhance the efficacy of T1DM management. In the current study, the factors associated with a higher DEPS-R score in patients with T1DM hence a higher risk of developing diabulimia include increased HbA1c levels, female sex, patients perceiving their weight higher than normal, not having received diabetes nutritional education, skipping breakfast, and having a first relative diagnosis of an eating disorder. Some of these factors could be modified and for this reason, it is valuable the future studies focus on assessing the effectiveness of their modification on diabulimia risk.

**Author Contributions:** Conceptualization, K.A. and F.G.; methodology, K.A. and F.G.; software, K.A.; validation, K.A. and F.G.; formal analysis, K.A.; investigation, K.A.; resources, K.A.; data curation, K.A.; writing—original draft preparation, K.A., M.R. and H.A.; writing—review and editing, K.A., M.R., H.A. and F.G.; visualization, K.A.; supervision, F.G.; project administration, K.A. and F.G.; funding acquisition, not applicable. All authors have read and agreed to the published version of the manuscript.

**Funding:** This research received no external funding.

**Institutional Review Board Statement:** The study was conducted in accordance with the Declaration of Helsinki, and approved by the the Ethics Committee of the "Attikon" General University Hospital, Athens, Greece. (approval code: ΒΙΠΠΚ ΕΒΔ245/20 April 2018).

**Informed Consent Statement:** Informed consent was obtained from all subjects involved in the study.

**Data Availability Statement:** The data presented in this study are available on request from the corresponding author. The data are not publicly available due to privacy issues.

**Acknowledgments:** The authors would like to thank all the staff of the Attikon and Laikon University Hospitals for the cooperation.

**Conflicts of Interest:** The authors declare no conflict of interest.

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
