# Peer review of "Factors Associated with Risk of Diabulimia in Greek Adult Population with Type 1 Diabetes Mellitus"

_2674-0311, doi:10.3390/dietetics2010003_

Round 1

Reviewer 1 Report

Major concerns:

- The authors mentioned about using a questionnaire in the methods section (line 78-92), was it a validated questionnaire? If yes, how it was validated?

- Self-reported weight and height for online participants should be listed as a limitation.

-  Line 165: Just because there was a significant association in males, it doesn’t mean that the reverse was correct for female, unless you have data to back it up.

- Other than male vs. female, the authors may also want to analyze the data across age groups.

- Linear regression analysis may not be appropriate for all the type of data that the authors have since most of the data are nominal and ordinal variables.

Minor Concerns:

- Provide reference for line 41

- Add 'individuals' after T1DM when referring to individuals with T1DM

- Exclusion criteria are not the 'opposite' of inclusion criteria

- Misspelled 'weighing', not weighting

- Line 211: ‘meta-analysis’, not metanalysis

- Line 243: Explain how insulin pump vs CGM may have introduced bias in this study.

- 'inversely' not inversingly in line 147

- There was an error with Table 1, for the row about 'Type of insulin administration'.

Author Response

Dear Editor,

The authors would like to thank all the reviewer for the recommended revisions and their invaluable assistance which have assisted us to significantly improve our article. 

Reviewer 1:

Comments and Suggestions for Authors

The authors would like to thank the reviewer for the thorough and helpful feedback, as well as the recommended revisions, all of which have been taken into consideration and have helped improve our article. 

Major concerns:

- The authors mentioned about using a questionnaire in the methods section (line 78-92), was it a validated questionnaire? If yes, how it was validated?

Response:

We would like to thank the reviewer for this observation. There is a short overview about the questionnaires used, all of which were validated (lines 107-121). The data presented in the lines 83-97 was collected through the common questionnaires used for demographic data and the simple questions about diet and lifestyle habits as stated in the manuscript. We have not used an extended questionnaire reflecting diet intake (eg. FFQ), exercise or sedentary behaviour. For this reason, it was not possible for us to investigate any other associations since we did not have an accurate assessment of participants’ nutritional status from these simple questionnaires, and thus we could not control the models for other important factors e.g. total energy intake etc. This observation was added in the limitations (lines 264-272)

- Self-reported weight and height for online participants should be listed as a limitation.

Response:

We would like to thank the reviewer for this comment. The limitation of all the self-reported data was added in the manuscript (lines 264-272).

-  Line 165: Just because there was a significant association in males, it doesn’t mean that the reverse was correct for female, unless you have data to back it up.

Response:

We appreciate this comment which gave us the chance to revise the expression used (line 20, 162) in order to reflect that indeed the association is significant in both sexes, but men had lower mean scores in DEPS.

- Other than male vs. female, the authors may also want to analyze the data across age groups.

Response:

In the context of stratified analysis, we have tested the interaction between sex and between age (as a categorical based on median and on tertiles), but the coefficient of the interaction was not significant with any of the other covariate used, Thus, we could not proceed to stratified analysis neither by sex, nor by age. This piece of information was added in line 154.

- Linear regression analysis may not be appropriate for all the type of data that the authors have since most of the data are nominal and ordinal variables.

Response:

We would like to thank the reviewer for this comment, which gave us the possibility to explain in detail the method used and the concept behind the analysis. Our dependent variable (DEPS score) is indeed a variable derived by the DEPS score which is a 16 item questionnaire with answers on a 6-point Likert (0 = never; 1 = rarely; 2 = sometimes; 3 = often; 4 = usually; 5 = always) and range from 0 to 80 points. Thus, it could be treated as a continuous variable, and use linear regression instead of ordinal regression. Regarding the predictors, it is true that the majority of them are categorical/ ordinal, but many are continuous variables eg age, Walking (min/week) etc. From the mathematical point of view, linear regression and ANOVA are identical: both break down the total variance of the data into different “portions” and verify the equality of these “sub-variances” by means of a test (“F” Test). In both techniques the dependent variable is a continuous one, but in the ANOVA analysis the independent variable(s) can be exclusively categorical variable, while in the regression can be used both categorical and continuous independent variables. Thus, ANOVA can be considered as a case of a linear regression in which all predictors are categorical, which is not our case since walking is continuous variable and if treated as categorical leads to loss of information. In addition, the four assumptions associated with a linear regression model (Linearity, homoscedasticity, independence of observations, normal distribution for Y based on any fixed value of X) were considered to be satisfactory in our selected model, as tested by the diagnostic plots (e.g.plot of residuals versus fitted values). For the analysis we have used the R studio which, in contrast to other statistic programs, allows to fully explore the collected data taking advantage of any statistical tool better describes the dataset.

Minor Concerns:

- Provide reference for line 41

Response:

As suggested, in line 41 references were added.

- Add 'individuals' after T1DM when referring to individuals with T1DM

Response:

As proposed, the word individuals was used after T1Dm, when necessary.

- Exclusion criteria are not the 'opposite' of inclusion criteria.

Response:

Exclusion criteria were revised accordingly to the reviewer’s comment. 

- Misspelled 'weighing', not weighting

Response:

The typo was corrected throughout the manuscript.

- Line 211: ‘meta-analysis’, not metanalysis

Response:

The typo was corrected.

- Line 243: Explain how insulin pump vs CGM may have introduced bias in this study.

Response:

We would like to clarify that for the purpose of the current study, insulin pump use was compared to multiple injections in the variable for the type of insulin administration. Diabulimia is a relative new eating disorder, which is not currently given a specific diagnostic code in the Diagnostic and Statistical Manual of Mental Disorders, Fifth Edition (DSM-5). To our knowledge, no previous research has identified possible triggering factors. For this reason, we assumed that the factors involved in diabetes mellitus management in addition to other individual’s characteristic, could possibly play a protective or an aggravating role in development of diabulimia. Thus, the hypothesis of the current study was to identify potential factors that are associated with DESP score.

- 'inversely' not inversingly in line 147

The typo was corrected.

- There was an error with Table 1, for the row about 'Type of insulin administration'.

Response:

Table 1 was corrected since Insulin Pump Users were the 65% and individuals on multiple injections were the 35% of the sample

Yours sincerely,

Kyriaki Apergi

Reviewer 2 Report

This cross sectional cohort study examines the important issue of "diabulimia" and identification of individuals at risk of this. The quality of english (grammar, spelling and phrasing) and data presentation (better tables) could be improved. The dietician effect is difficult to understand- page 3 line 135 (I don't like the dietician supervision) vs page 8 line 184- nutrition therapy delivered by a registered dietitian was associated with absolute decreases in HbA1c.

A comment as to the advantages of the EAT-26 compared with the eating disorder examination questionnaire would be useful

The study is worthwhile in raising clinician awareness of the possibility of eating disorders in their patients as a possible reason for poor control and clinical indicators that point to the possibility of this.

Author Response

Dear Editor,

The authors would like to thank all the reviewers for the recommended revisions and their invaluable assistance which have assisted us to significantly improve our article. 

Reviewer 2

Comments and Suggestions for Authors

This cross sectional cohort study examines the important issue of "diabulimia" and identification of individuals at risk of this. The quality of English (grammar, spelling and phrasing) and data presentation (better tables) could be improved. The dietician effect is difficult to understand- page 3 line 135 (I don't like the dietician supervision) vs page 8 line 184- nutrition therapy delivered by a registered dietitian was associated with absolute decreases in HbA1c.

A comment as to the advantages of the EAT-26 compared with the eating disorder examination questionnaire would be useful

The study is worthwhile in raising clinician awareness of the possibility of eating disorders in their patients as a possible reason for poor control and clinical indicators that point to the possibility of this.

Response:

We would like to thank the reviewer for the comments. The manuscript was revised by a native English speaker and we tried to organised better the tables. Line 135 (now 136) was corrected. Line 184 refers to literature and not to our results.

As for the comment regarding the use of EAT-26 instead of Eating Disorder Examination Questionnaire (EDE-Q), in the current study, we prefer to use EAT-26, as a validated tool that is frequently used by the Greek healthcare practitioners.

Yours sincerely,

Kyriaki Apergi

Reviewer 3 Report

Reviewer                                           Prof. Dr. med. Peter E.H. Schwarz

Title                                                      Factors Asscociated With Risk Of Diabulimia In Greek Adult Population With Type 1 Diabetes Mellitus

General comments                        T1DM patients are particularly at risk of developing an eating disorder. In recent studies, 60% of T1DM patients admit to misusing insulin, which leads in noticeably earlier occurence of diabetic complications. Consequently, the present study addresses a highly relevant clinical topic, both from the perspective of long-term quality of life and from a cost aspect for the healthcare system.

The overall quality of the work is very high. The data analysis is clear and well done. Appropriate methods were used. A minor revision is necessary.

Comments on format                    The manuscript is very clearly structured. It has a common thread and is coherent in itself. In chapter 2, a new paragraph should be added after the sentence of exclusion criteria.

Comments on language               Spelling and grammar is good. Some wording could be improved. The manuscript has some minor typos.

Comments on references            Sufficient references were used to clearly prove made statements. Only in line 41 a reference should be added. The references are up to date, in line 116 a more recent one could possibly be used. All references were correctly cited and listed in the bibliography.

Comments on methods               Description of the methodology is well done. Especially the given short overview about the questainnaires is worthy. Questionnaires used were validated ones, suited for the study. The DEPS-R is considered as a very reliable instrument with a high α for internal consistency. As the norm-scores are dependent on age, sex and age- / sex-specific BMI, these parameters were correctly taken into account by the authors. The BDI-II also has a good repeatability and is a high-quality instrument. Perhaps instead of the EAT-26, the more economical, shortened version of the EAT-13 might have been sufficient.  

The reviewer wonders if the study population of T1DM patients also includes the group of LADA (late onset autoimmune diabetes in adults) and if so, whether a group-specific analysis would have been beneficial. The reviewer assums that LADA, who were not treated with insulin from the beginning of their disease, act differently in insulin therapy.

Approval of Ethics Commitee with reference was reported.

Comments on Results                   The data presentation is well structured and the results are plausible.

Selected tests for statistical analysis were correctly chosen and performed. Not sure if possibly the use of beta coefficients instead of unstandardised b coefficients would have been more powerful regarding estimation of the relative impact of each predictor to the overall prediction.

In the legend of tables 1 and 2, I would use the official term "self-monitoring blood glucose" for SMBG. There is a typing mistake in the legend of table 2 with regard to CGM. BGC (blood glucose control) is not mentioned in the table.

Comments on Discussion            The discussion section is particularly well done. The authors have presented in detail the context with the current state of studies and literature. Limitations and possible bias were mentioned, for example the use of HbA1c as an average value. Perhaps TIR (time-in-range) values of the included CGM-users can provide more explicit results in a subsequent data analysis. Risk of bias due to predominantly female sex among the patients was stated. Moreover it should also be mentioned that information about body weight was only collected via self-reporting. This is also applicable to other self-reported data given, such as organized exercise and walking time, as well as dietary patterns or if a first relative diagnosed with eating disorder.

Comments on Conclusions         Factor of time spent walking per week is missing, see abstract and results.

Author Response

Dear Editor,

The authors would like to thank all the reviewers for the recommended revisions and their invaluable assistance which have assisted us to significantly improve our article. 

Reviewer 3:

Prof. Dr. med. Peter E.H. Schwarz

Title                                                      

Factors Asscociated With Risk Of Diabulimia In Greek Adult Population With Type 1 Diabetes Mellitus 

General comments                        

T1DM patients are particularly at risk of developing an eating disorder. In recent studies, 60% of T1DM patients admit to misusing insulin, which leads in noticeably earlier occurrence of diabetic complications. Consequently, the present study addresses a highly relevant clinical topic, both from the perspective of long-term quality of life and from a cost aspect for the healthcare system.

The overall quality of the work is very high. The data analysis is clear and well done. Appropriate methods were used. A minor revision is necessary.

Response:

We would particularly appreciate the time and the effort the reviewer spend in order to assist us improve the quality of our manuscript.

Comments on format                    

The manuscript is very clearly structured. It has a common thread and is coherent in itself. In chapter 2, a new paragraph should be added after the sentence of exclusion criteria.

Response:

As suggested, in chapter 2 (line 79), a new paragraph was added after the sentence of exclusion criteria.

Comments on language               

Spelling and grammar is good. Some wording could be improved. The manuscript has some minor typos.

Response:

We would like to thank the reviewer for this comment we had a native English speaker to check all the manuscript, including correction of typos.                                                           

Comments on references           

Sufficient references were used to clearly prove made statements. Only in line 41 a reference should be added. The references are up to date, in line 116 a more recent one could possibly be used. All references were correctly cited and listed in the bibliography.

Response:

As suggested, in line 41 references were added and in line 116, we prefer to keep the reference 17 (Beck At, et al 1961), because it reflects to the initial validation of the BDI instrument, but we also added a reference about the Greek validation and use of the instrument.

Comments on methods               

Description of the methodology is well done. Especially the given short overview about the questionnaires is worthy. Questionnaires used were validated ones, suited for the study. The DEPS-R is considered as a very reliable instrument with a high α for internal consistency. As the norm-scores are dependent on age, sex and age- / sex-specific BMI, these parameters were correctly taken into account by the authors. The BDI-II also has a good repeatability and is a high-quality instrument. Perhaps instead of the EAT-26, the more economical, shortened version of the EAT-13 might have been sufficient.  

Response:

We particularly appreciate this comment, giving us the chance to provide a detailed explanation of the instruments used. In the current study, we prefer to use EAT-26, as a validated tool that is frequently used by the Greek healthcare practitioners. Although EAT-13 might be more economical and shortened than EAT-26, it was not validated neither in the Greek population, nor in patients with type 1 diabetes.

The reviewer wonders if the study population of T1DM patients also includes the group of LADA (late onset autoimmune diabetes in adults) and if so, whether a group-specific analysis would have been beneficial. The reviewer assums that LADA, who were not treated with insulin from the beginning of their disease, act differently in insulin therapy.

Approval of Ethics Commitee with reference was reported.

Response:

In the context of the reviewer’s comment, the current sample (n=100) did not included any LADA patients. Thus, we have added a sentence in line 136-137 in order to reflect this very important remark.

Comments on Results                   

The data presentation is well structured and the results are plausible.

Selected tests for statistical analysis were correctly chosen and performed. Not sure if possibly the use of beta coefficients instead of unstandardised b coefficients would have been more powerful regarding estimation of the relative impact of each predictor to the overall prediction.

Response:

According to the reviewer’s observation, we would like to mention that we have prefer to use the unstandardised b coefficients in order to reflect to each variable change in their original scales i.e., in the same units in which we are taken the dataset from the source to train the model. We believe that this will help the reader to better understand the significance of each variable, since each unstandardised b coefficient is interpretated is straightforward and intuitive: if all the other variables in the model are held constant, a shift of 1 unit in X(predictor variable) implies there is an average change of b units in Y(outcome). Thus, unstandardized coefficients are considered to be ideal for interpreting the relationship between an independent variable X (predictor) and an outcome Y. However, they are not useful for comparing the effect of an independent variable with another one in the model, and if it is needed we could additionally provide the standardised beta coefficients.

In the legend of tables 1 and 2, I would use the official term "self-monitoring blood glucose" for SMBG. There is a typing mistake in the legend of table 2 with regard to CGM. BGC (blood glucose control) is not mentioned in the table.

Response:

According to the reviewer’s observation, we would like to mention that we appreciate reviewer’s note and we have corrected the term with the official "self-monitoring blood glucose" for SMBG in the legend of tables 1(line 142) and 2 (line 153) and the typo in legend of the 2nd table (line 152).

Comments on Discussion            

The discussion section is particularly well done. The authors have presented in detail the context with the current state of studies and literature. Limitations and possible bias were mentioned, for example the use of HbA1c as an average value. Perhaps TIR (time-in-range) values of the included CGM-users can provide more explicit results in a subsequent data analysis. Risk of bias due to predominantly female sex among the patients was stated. Moreover it should also be mentioned that information about body weight was only collected via self-reporting. This is also applicable to other self-reported data given, such as organized exercise and walking time, as well as dietary patterns or if a first relative diagnosed with eating disorder.

Response:

We appreciate this note and we have added a sentence (line 274-276) reflecting the self-reported nature of the instruments used.

Comments on Conclusions         

Factor of time spent walking per week is missing, see abstract and results.

Response:

The factor of walking per week is measured in minutes and it was corrected.

Yours sincerely,

Kyriaki Apergi

Round 2

Reviewer 1 Report

The authors have responded to all critiques. Minor edits are needed.

It appears that the authors have addressed all the major concerns which were brought up to their attention in the past.

Minor concerns:

Page/Line/Table/Figure

Remarks

Line 53

… influence on T1DM…

Line 53

… not yet recognized as what?

Line 58

… may lead to…

Line 62

Hence, the purpose…

Line 83 - 97

Close your parentheses before giving new examples.

Line 123

The analyses

Line 124

Descriptive statistics were provided…

Line 136

Table 1

Line 154

No statistically significant interaction…

Table 3

Bold: Statistically significant (p < 0.05)

Line 247

When controlled for covariates.

Author Response

Dear Editor,

Dear Reviewer,

We would like to thank the Reviewer for taking their time to review once more our manuscript and appreciate the constructive and thoughtful corrections, all of which have been taken into consinderation. All changes are marked in revised manuscript. Please find below our point-by-point responses to the Reviewer’s comments and suggestions.

Page/Line/Table/Figure

Remarks

Action taken

Line 53

… influence on T1DM…

corrected

Line 53

… not yet recognized as what?

diabulimia is not yet recognized as a mental illness

Line 58

… may lead to…

corrected

Line 62

Hence, the purpose…

corrected

Line 83 - 97

Close your parentheses before giving new examples.

corrected

Line 123

The analyses

corrected

Line 124

Descriptive statistics were provided…

corrected

Line 136

Table 1

corrected

Line 154

No statistically significant interaction…

corrected

Table 3

Bold: Statistically significant (p < 0.05)

corrected

Line 247

When controlled for covariates.

corrected

Dr.  Kyriaki Apergi
